# Hyperspectral Image Data and Waveband Indexing Methods to Estimate Nutrient Concentration on Lettuce (*Lactuca sativa* L.) Cultivars

**DOI:** 10.3390/s22218158

**Published:** 2022-10-25

**Authors:** Sulaymon Eshkabilov, John Stenger, Elizabeth N. Knutson, Erdem Küçüktopcu, Halis Simsek, Chiwon W. Lee

**Affiliations:** 1Department of Agricultural and Biosystems Engineering, North Dakota State University, Fargo, ND 58108, USA; 2Department of Plant Sciences, North Dakota State University, Fargo, ND 58108, USA; 3Department of Agricultural Structures and Irrigation, Ondokuz Mayıs University, Samsun 55139, Turkey; 4Department of Agricultural and Biological Engineering, Purdue University, West Lafayette, IN 47907, USA

**Keywords:** lettuce, hyper-spectral imaging, nutrient level, reflectance, spectral index, partial least squares regression, hydroponic culture

## Abstract

Lettuce is an important vegetable in the human diet and is commonly consumed for salad. It is a source of vitamin A, which plays a vital role in human health. Improvements in lettuce production will be needed to ensure a stable and economically available supply in the future. The influence of nitrogen (N), phosphorus (P), and potassium (K) compounds on the growth dynamics of four hydroponically grown lettuce (*Lactuca sativa* L.) cultivars (Black Seeded Simpson, Parris Island, Rex RZ, and Tacitus) in tubs and in a nutrient film technique (NFT) system were studied. Hyperspectral images (HSI) were captured at plant harvest. Models developed from the HSI data were used to estimate nutrient levels of leaf tissues by employing principal component analysis (PCA), partial least squares regression (PLSR), multivariate regression, and variable importance projection (VIP) methods. The optimal wavebands were found in six regions, including 390.57–438.02, 497–550, 551–600, 681.34–774, 802–821, and 822–838 nm for tub-grown lettuces and four regions, namely 390.57–438.02, 497–550, 551–600, and 681.34–774 nm for NFT-system-grown lettuces. These fitted models’ levels showed high accuracy $\left(R^{2}=0.85-0.99\right)$ in estimating the growth dynamics of the studied lettuce cultivars in terms of nutrient content. HSI data of the lettuce leaves and applied N solutions demonstrated a direct positive correlation with an accuracy of 0.82–0.99 for blue and green regions in 400–575 nm wavebands. The results proved that, in most of the tested multivariate regression models, HSI data of freshly cut leaves correlated well with laboratory-measured data.

## 1. Introduction

Lettuce (*Lactuca sativa* L.) is one of the most important fresh leafy vegetables. It is often consumed raw in green salads. The quality, yield, and health of lettuce are directly related to the availability of mineral nutrition through adequate fertilization. Optimal nutrient supply improves quality and growth parameters, such as plant height, total biomass, leaf number, leaf size, protein and chlorophyll content, antioxidant capacity, flavor, color, and dry matter quantity [1]. Nitrogen (N) is one of the most important nutrients that promotes growth and influences the phytochemical response of plants [2]. Deficiency of N is a major abiotic growth-limiting factor for lettuce production. It affects leaf length, root-to-leaf ratio, leaf fresh weight, and root biomass and also reduces the accumulation of ascorbic acid and flavonoids in lettuce leaves [3]. However, excess N also causes undesirable consequences for the plant and the environment [4,5]. Therefore, N management and monitoring of N content in leaves are critical for efficient and sustainable production. Determination of lettuce leaf N concentration in the laboratory is costly and time-consuming and requires special tools and trained specialists. Therefore, current advances in plant phenotyping technology must be used to reduce the cost of nutrient analysis of plant leaves [6,7,8].

Two standard methods are generally used to evaluate the N concentration of plants, namely destructive and non-destructive methods. Traditional methods are destructive, involving laboratory-based measurement of N concentration on tissue samples. These methods are labor intensive and time-consuming because of their requirement for manual collection of plant leaves. Destructive methods, in addition, may interfere with other measurements or experiments because of the influence leaf removal has on the plants. Researchers proved that N status in plants is directly linked to the formation of chlorophyll [9,10]. Therefore, chlorophyll content in leaves can theoretically be considered as an indicator of N status [11]. The soil and plant analyzer development (SPAD) reading provides an indirect value reflecting chlorophyll content [12]. This reading compares light absorption at a wavelength associated with chlorophyll (such as 650 nm) with that not associated (such as 940) [13]. Through this method, a simple, non-destructive test can be conducted to gain insight into plant nitrogen status. However, Xiong et al. [14] reported on the influence environmental conditions has on the accuracy of SPAD readings relative to N fertilization. They found readings were closely associated with chlorophyll content; however, N content varied widely in comparison with either chlorophyll content or SPAD reading. Inconsistency of obtained values can lead to inaccurate grower responses and mismanagement of crop production. Additionally, the exact chlorophyll concentration of the entire leaf cannot be determined by this method because it only provides data on the chlorophyll content of the leaves at a single point in time and a small sampled area relative to the plant.

Non-destructive methods, on the other hand, are simpler, faster, less expensive, and less labor-intensive than destructive methods and can determine N concentration without harming the plants [15]. Hyperspectral imaging (HSI), a non-destructive method, has increased in importance in recent years because of advances in computer vision and digital imaging technologies [16]. HSI combines the advantages of spectroscopy with those of digital imagery to provide a comprehensive overview of an object’s physical, chemical, and biological properties. Hence, this technique has experienced tremendous growth and has been used with great success in agriculture, including the detection of water status, biomass, yield, nutrient status, and disease control. Sun et al. [17] performed an analysis of HSI to estimate the amount of water present in corn leaves. Wang et al. [18] used the HSI and a chemometric approach to estimate K and sodium contents in tea leaves. The best correlation coefficients for the estimation (0.9423 and 0.9168 for P and K, respectively) were obtained by combining the sequential projection algorithm with multiple linear regression models. In a field study, Wang et al. [19] evaluated HSI in combination with chemometric methods for qualitative and quantitative diagnosis of N status of tea plants and obtained a correlation coefficient of 0.924 for estimating N content of the leaves. Sabzi et al. [20] applied HSI to determine the N content of cucumber (*Cucumis sativus* L.) leaves from plants receiving various fertilizer-application treatments. Either joint prediction or individual models were applied depending on the significant differences between treatments. Finally, three different regression models were used to predict N content of the cucumber leaves. Such models successfully aided in the prediction of N content of cucumber using data from HSI. The study’s outcomes might help farmers to apply an optimum amount of chemical fertilizer in the field to proactively prevent overapplication and its many negative impacts on the plant, production efficiency, and the environment. In another study, Eshkabilov et al. [16] captured HSI of freshly cut lettuce leaves with a hyperspectral camera and developed algorithms to determine nitrate (NO^3−^), potassium (K^+^), calcium (Ca^2+^), soluble solids, pH, and total chlorophyll concentrations in different lettuce varieties. The results showed that these parameters could be estimated in the 400 to 1000 nm range using HSI processing techniques. Nutrient concentrations determined by chemical methods in the laboratory were well-correlated with least squares principal component analysis (PCA) techniques of the four optimal wave bands. Non-destructive spectral indices were used to predict the carotenoid and anthocyanin content of various lettuce cultivars grown under different conditions, including organic, hydroponic, and alternative growing systems. Several mathematical models were applied to estimate lettuce pigments. Exponential and linear fit models provided the best estimation of lettuce pigments [21].

It is critical to properly select the region of interest (ROI) in the HSI when processing data. The spectral characteristics of an object under study are determined using ROI in a HSI by calculating the average of the individual pixels. Digital filters, such as Savitzky-Golay or moving averages, are often used in the calculation of average pixel values [22]. In addition, various statistical analyses are applied to HSI data, such as PCA, standard deviation, nearest neighbor, cross-correlation, and partial least squares regression (PLSR). Zhan-qi et al. [23] used PCA to distinguish spinach leaves with different dimethoate pesticide residue levels based on this technique. Steidle et al. [24] developed a PLSR model to predict the chlorophyll, carotenoid, and anthocyanin content of lettuce. Yu et al. [13] employed a PLSR model to successfully predict the total N content of the whole plant (leaf-stem-root). Asante et al. [25] studied the effects of freeze damage treatment on N content in tea leaves using HSI and analyzed them with PCA, PLSR, and linear models. Jin et al. [26] determined the water content in peanut kernels by HSI technique and constructed a quantitative PLSR model with an optimal coefficient of determination of 0.91.

The above literature has contributed significantly to our understanding; however, no previous study has adequately evaluated the relationship between nutrient concentration with yield and quality parameters of lettuce in different cropping systems. Therefore, this study was intended to reveal the relationship between the amount of N applied with growth dynamics, response, and quality of lettuces grown in tub and nutrient film technique (NFT) systems. Similarly, there exists little information on employing HSI, linear multivariate regression modeling, and statistical methods to estimate nutrient content on freshly cut leaves of different lettuce cultivars.

To address these gaps in the literature, this paper integrated the HSI technique into a linear multivariate regression model to estimate nutrient content on fresh-cut leaves of four lettuce cultivars grown in hydroponic tubs and in the NFT system at different N, K, and P solution levels. The novelty of the study is the developed methodology to estimate the growth dynamics of lettuce cultivars using HSI data and three different waveband-indexing methods. A correlation between HSI scanning of the nutrient solution versus HSI scanning of lettuce leaves was, for the first time, studied, and a good correlation between nutrient concentrations in the solution and the leaves was found. 

## 2. Materials and Methods

### 2.1. The Influence of N Concentration on Lettuce Growth Dynamics

Four different lettuce cultivars, including Black Seeded Simpson (Johnny’s Selected Seeds, ME, USA), Parris Island Cos—Romaine Lettuce (Eden Brothers, NC, USA), Rex (Rijk Zwaan, Salinas, CA, USA), and Tacitus (Rijk Zwaan, Salinas, CA, USA), were grown in a greenhouse at 16-h day/8-h night with the light intensity of 200–300 µmol·m^−2^·s^−1^ and temperatures of 70 °F day/60 °F night. The four cultivars were grown: (i) in hydroponic tubs using 8 different nutrient concentrations (0, 50, 100, 150, 200, 250, 300, and 350 ppm N) of a commercial 20-20-20 fertilizer (N-P_2_O_5_-K_2_O, Jack’s Fertilizer, Allentown, PA, USA) and 0, 22, 44, 66, 88, 110, 132, and 154 ppm P concentrations and 0, 41.5, 83, 124.5, 166, 207.5 249, and 290.5 ppm K concentrations and (ii) in an NFT recirculation system using four different N concentrations (50, 100, 200, and 400 ppm N concentrations) and 22, 44, 88, and 167 ppm P concentrations and 41.5, 83.0, 166, and 332 ppm K concentrations. Four samples of each lettuce cultivar were grown at each concentration. The seeds of the four cultivars were germinated on rockwool cubes (1.0 × 1.0 × 1.5 inches) in plastic trays, and seedlings were grown for two weeks with nutrient solution (200 ppm N). The two-week-old seedlings were transplanted to a Styrofoam board (6 plants/board), which was floated on top of nutrient solution contained in 12 qt (11.35 L) plastic hydroponic tubs (Figure 1a,b). Similarly, two-week-old seedlings were transplanted onto an NFT circulation system (Figure 1c,d). 

The nutrient solution in tubs (10 L/tub) was gently aerated with an air-stone diffuser connected to compressed air through silicone tubing. In both systems, N concentration solutions were reloaded each week for three weeks until the harvesting day. Four randomly selected fresh leaves of all harvested lettuce cultivars were studied in the laboratory. Leaf sample nutrient concentrations, including NO^3−^, K^+^, and Ca^2+^, were measured (Model: S030, S040, S050, LAQUAtwin—NO3-11, K-11, CA-11, Horiba Scientific, Kyoto, Japan). The potential hydrogen (pH) levels in leaves were measured using a pH meter (Model: S010, pH-Horiba, Horiba Scientific, Kyoto, Japan). Tissue sugar content (Brix, %) was measured using a digital refractometer (Refractometer PAL-1, Atago Inc., Tokyo, Japan). Relative concentration of chlorophyll (SPAD reading) levels of leaves was measured with SPAD-502 Plus (Konica Minolta Sensing, Tokyo, Japan). SPAD readings are related to greenness by transmitting light from a light diode through a leaf at wavelength 650 nm to 940 nm, with the 650 nm light a peak chlorophyll attenuating of red light. In addition, fresh and dried leaf weights (g) were measured. Ten sampled readings from freshly leaf samples from each lettuce cultivar were collected and statistically treated as subsamples of the experimental unit by computing their average values and standard deviations. 

### 2.2. HSI Capture

HSI of freshly cut, randomly selected four leaf samples of each lettuce species grown in a hydroponic tub and NFT system were acquired using a push-broom scanning hyperspectral camera (Pika XC2, Resonon Inc, Bozeman, MT, USA) with 1600 spatial pixels and 462 spectral bands ranging between 390.57–1008.6 nm with fixed Xenoplan lens (1.4/23-0902, Schneider Kreuznach, Stockach, Germany) with a focal length of 23 mm and 23.10 with IFOV of 0.52 mrad (Figure 2a). The HSI system was installed in a closed dark chamber (Figure 2b). The HSI unit has a built-in self-correction for white and dark references. All captured HSIs were acquired via a laptop using Spectronon 3 software provided with the camera.

### 2.3. HSI Indices

Reflectance values of the captured HSI data were processed using the MATLAB package and its image processing and digital signal toolboxes (MATLAB 2022a, MathWorks, Inc., Natick, MA, USA). Before processing, the acquired HSI data were filtered using a 13-point moving average filter. The first-order derivative of the reflectance (FDR) data was computed from the normalized difference spectral (NDS) of the filtered HSI data according to the formulations given in [16]. The FDR values clearly demonstrated several bandwidth regions in NDS data where significant dynamics of reflectance values were present. Using these highly dynamic regions of the reflectance values, two different bands, such as high band- and low band-reflectance values, were chosen to find two indices [27].

### 2.4. Feature Extraction Models

To extract features from the acquired HSI reflectance values, two multivariate prediction models, namely principal component analysis (PCA) [28] and partial least squares (PLSR) [29] modeling approaches, were performed. The PCA and PLSR models are employed in reducing high-dimensional multivariate problems and ill-condition datasets [30] and solving various multivariate problems [31,32]. The PLSR model is expressed by Equations (1) and (2) [33].
(1)$$X=TP^{T}+\tilde{X}.$$
(2)$$Y=UQ^{T}+\tilde{Y}.$$
where *X* and *Y* are representing predictor and response matrices, whose sizes are n × p and n × 1*,* decomposed into score matrices *T* and *U* of size n×h. Matrices *P* and *Q* of size p×h and h×1 are loading matrices, and $\tilde{X}$ and $\tilde{Y}$ are randomly distributed error matrices. Predictor matrix *X* is the computed average reflectance values, and response matrix (*Y*) represents the nutrient values measured via laboratory measurements. The YX-PCA model Z = [Y X] was obtained using Equation (3) proposed by Yue and Qin [34].
(3)$$Z=T_{z}P_{z}+\tilde{Z}.$$
where $T_{z}\;\mathrm{and}$ $P_{z}$ are score matrices of size n×h and n×1. $\tilde{Z}$ is the randomly distributed residual error. The found multivariate PLSR and PCA models based on the computed reflectance values of the HSI data were used to predict NO^3−^, Ca^2+^, and K^+^ in ppm; Brix in %; pH- and SPAD-applied N in ppm; and fresh and dry leaf weight (g) of freshly cut leaves. The predicted values of nutrients were computed using the multivariate regression formula given in Equation (4) [35].
(4)$$Y=\;\sum_{i=1}^{n}\beta_{i}\;H_{i}+C$$
where $\beta_{i}$ is the fit model coefficient (regression coefficient) of the PLSR and PCA models; $H_{i}$ is the spectrum of each pixel in the HSI, and *C* is the constant term. 

The accuracy of found response variable (*Y, Z*) values was validated by computing the correlation coefficient of determination (R^2^) and root mean square errors (RMSE) from Equation (5).
(5)$$\mathrm{RMSE}=\;\sqrt{\frac{\sum_{k=1}^{n}{\left(y_{k}-y_{p}\right)}^{2}}{n}\;}.$$
where *y_k_* and *y_p_* are the measured (in the laboratory) and predicted values of NO^3−^, Ca^2+^, K^+^, Brix, pH, SPAD, N, and leaf weight for sample k, respectively, and *n* is the number of samples in the data set. Cross-correlation between the HSI data and applied N concentration solutions were computed using the statistical cross-correlation formula given in Equation (6).
(6)$$r_{corr}=\frac{\sum_{i=1}^{n}(\left(x-\bar{x}\right)\left(y-\bar{y}\right))}{\sqrt{\sum_{i=1}^{n}{\left(x-\bar{x}\right)}^{2}\sum_{i=1}^{n}{\left(y-\bar{y}\right)}^{2}}}.$$
where x is a predictor (input) variable taken from HSI reflectance data of the applied N concentration solutions, and $\bar{x}$ is its mean value. y is a response variable from HSI reflectance data of the freshly cut lettuce leaves, and $\bar{y}$ is its mean value.

### 2.5. PLS-VIP Method

The VIP score of a predictor variable is a high-quality estimator and selector of more important predictors for the projections to locate latent variable *h* values. The VIP score for *j*th variable can be computed using Equation (7) [36]. The mean value of the VIP score is “1”; thus, the best score values will be higher than 1, and the least important scores will be below “1”.
(7)$$VIP_{j}=\;\sqrt{p\;\overset{h}{\underset{i=1}{\sum }}\left(SS\left(\beta_{i}t_{i}\right){\left(\frac{w_{j,i}}{\left\|W_{j}\right\|}\right)}^{2}\right)/\overset{h}{\underset{i=1}{\sum }}SS\left(\beta_{i}t_{i}\right)}\;.$$
where $SS\left(\beta_{i}t_{i}\right)=\beta_{i}^{2}t_{i}^{T}t_{i},\;$and $\beta_{i},\;t_{i}$ stand for the *i*th column of matrices *T* and *P* defined in Equations (1) and (3). The parameter *p* is a size (number) of a predictor variable $x_{i}$ to be used in the regression model. The vector $w_{j,\;i}$ is the *i*th element of the weight matrix $W_{j}$. Moreover, the score vector $t_{i}$ is computed from Equation (8) in relation to the weight vector and predictor variable $x_{i}$.
(8)$$t_{i}=W_{j}^{T}x_{i}.$$
where the predictor variable $x_{i}$ is the HSI’s averaged reflectance value corresponding to *i*th bandwidth.

### 2.6. Waveband Selection Methods

Three approaches were used to select the wave range index. First, the FDR values, as explained in Section 2.3, were computed by taking the first-order numerical derivative of the averaged (smoothed) reflectance values and plotted against waveband values. The plotted FDR values clearly showed six distinctive dynamic waveband regions for hydroponically grown lettuces and four regions for NFT-system-grown ones. Six waveband regions found with the FDR data from the HSI data of hydroponically grown lettuces were identified, and six waveband values were determined from these regions. Similarly, for the lettuces grown in the NFT system, four bandwidth regions were identified, and four waveband values were found. Then, multivariate regression model coefficients ($\beta_{i}$) were determined by using the found HSI reflectance values corresponding to the found wavebands used to predict the response variables. 

Second, the found regression model coefficients ($\beta_{i}$) from the PLSR and PCA and the taken reflectance values of the captured HSI reflectance data from all wavebands were used to predict the response variables. To avoid overfitting, the fittest wavebands of the captured HSI reflectance were found. The $\beta_{i}$ values that showed the HSI reflectance values from the wavelengths that had significantly higher importance than the others were assessed. In this way, the most important waveband values were located. Six essential HSI reflectance values were found corresponding to six wavebands, namely two wavebands from the blue, two wavebands from the green, and two from the red-color waveband region. 

Third, VIP-score-based waveband selection was implemented by computing VIP-score values using Equations (7) and (8). From the computed VIP scores and indices of the wavebands, those which exhibited higher than “1” scores were located. These values were then used to select respective HSI reflectance values. 

The quality and accuracy of the chosen wavebands and found models were assessed using the correlation coefficient of determination ($R^{2}$) and root mean square errors (RMSE) based on Equation (5). Moreover, the cross-correlation between the HSI data of the freshly cut lettuces grown in hydroponic tubs and nutrient solutions was calculated using Equation (6) to demonstrate the relationship between the amount of N applied and the nutrient content of the harvested lettuce leaves.

## 3. Results and Discussion

### 3.1. Hydroponic System

In general, all lettuce cultivars grown in hydroponic tubs with nutrient solution grew well, while those receiving only tap water did not (Table 1 and Table 2). The total biomass yield increased with N solution concentration increase, and leaf-edge burns occurred more frequently in plants grown at 300 to 350 ppm N (Table 1). It is a common practice to use 200 ppm N concentration in the nutrient solution for hydroponic-culture lettuce [6]. In an experiment by Odabas et al. [6], it was found that hydroponic solutions containing 100 to 150 ppm N provided excellent growth of lettuce without resulting in blemishes for all three cultivars tested. Maximum growth and weight of Black Seeded Simpson, Parris Island, Rex RZ, and Tacitus cultivars grown in hydroponic tubs were achieved in 250, 250, 100, and 200 ppm N treatments, respectively (Table 1). These results are similar to the research carried out by Sapkota et al. [37], who found that 200 ppm of N should be used to maximize lettuce growth for the hydroponic culture of the Buttercrunch cultivar. The N can be taken up by plants as either an anion (NO_3_^−^) or a cation (NH_4_⁺). When the NH_4_⁺ concentration was predominant under hydroponic conditions, the solution became very acidic, and plant growth was reduced [38]. These deleterious effects of NH₄⁺ may be one of the reasons for the restrained plant growth at certain N concentrations in the present study. In contrast, the content of NO_3_^−^ in leaf tissue increased with the increase of N concentration from 50 to 350 ppm. Sahin and Seckin [39] observed that lettuce yield was significantly affected by NO_3_^−^ content and that increasing the N dose increased plant NO_3_^−^ accumulation with increasing biomass. Stefanelli et al. [40] found that the accumulation of NO_3_^−^ was very high at levels of N over 1200 ppm but below the minimum acceptable level at levels of 40 to 400 ppm.

While the N concentration in the nutrient solution increased, the contents of K^+^ and Ca^2+^ in the leaf tissue remained relatively constant (Table 2). It is noteworthy that K^+^ in the leaf did not fluctuate because of increased K^+^ concentration in the nutrient solution, considering that as N concentration in the solution increased, so did the K^+^ in the nutrient solution. Lettuce plants must have the ability to maintain leaf tissue K^+^ content regardless of changes in K^+^ concentration in the nutrient solution. 

### 3.2. NFT System 

As nutrient concentration increased, leaf chlorophyll content as indicated by SPAD reading increased, while leaf tissue pH and Brix readings remained consistent (Table 3). It has been reported that the SPAD readings and N concentrations are highly correlated in a variety of plant species [41,42,43]. Sahin and Seckin [39] obtained SPAD readings of 5.7, 9.0, and 12.3 when N was applied at 50, 100, and 150 ppm, respectively, using the lettuce variety “Funnly F1” as material. The high Brix value observed on plants grown with tap water, without fertilizer addition, is noteworthy. Since these plants had little growth and struggled to survive, this finding is not considered significant growth. The lettuce cultivars cultured in the NFT nutrient-circulation system showed similar growth responses to those grown in hydroponic tubs. While all N concentrations of the solutions provided good growth, plants developed best with nutrient solutions containing 100 and 200 ppm N concentrations. Of all treatments, the visual quality of lettuce was the best when plants were grown with 100 ppm N in the nutrient solution and had the fewest leaf-edge burns. Like the hydroponic tub cultures, the lettuces grown with increasing N concentrations in the circulating nutrient solution in the NFT system did not have significantly different Brix readings for 50, 100, and 200 ppm N concentrations and leaf tissues pH values for all N concentrations (Table 3). 

### 3.3. HSI Capture

The collected HSI data show that all four lettuce cultivars had similar HSI reflectance value trends for 390.57 to 1008.57 nm wavebands except for the leaves from the lettuce cultivars grown in water (0 ppm N concentration) (Figure 3a–d). N concentration solutions (applied treatments) in hydroponically grown lettuce cultivars were also scanned with the hyperspectral camera with two repetitions, and the acquired HSI data were averaged (Figure 3e). To determine the correlation between the applied N concentrations in solutions and lettuce leaves, a cross-correlation analysis of averaged values of lettuce and solution HSI data was performed using Equation (6) within the blue and green bandwidth regions 390.57 to 554.48 nm.

The cross-correlation results in Table 4 clearly show that there is a statistically significant correlation between the HSI data of the lettuce leaves and the solutions for the 390.57 to 554.48 nm bandwidth region. The highest correlation values for Black Seeded Simpson, Parris, Rex RZ, and Tacitus were obtained at N concentrations of 300 ppm (r_corr_ = 91.8%), 350 ppm (r_corr_ = 93.6%), 350 ppm (r_corr_ = 95.2%), and 350 ppm (r_corr_ = 94.1%), respectively, except for those grown with tap water only (0 ppm N concentration). As with hydroponically grown lettuces, HSI images were collected from freshly cut leaves of the four cultivars grown in the NFT system (data not shown).

As mentioned in Section 2.6, wavebands were selected using three approaches. First, a first-order numerical derivative of the averaged (smoothed) reflectance values was performed and plotted against the waveband values. Figure 4a,c shows the computed 32 NDS data (provided in Appendix A) obtained from the acquired HSI images of lettuces grown in hydroponic tubs and the computed FDR values from the 16 NDS data, respectively. Similarly, Figure 4b,d shows the computed 16 NDS data obtained from the acquired HSI images of lettuces grown in NFT system and the computed FDR values from NDS data, respectively. The plotted FDR values clearly showed six distinctive dynamic waveband regions for hydroponically grown lettuce (Figure 4a–c) and four regions for lettuce cultured in the NFT system (Figure 4b–d). Region 1: 390.57 to 491.3 nm, region 2: 496 to 547 nm, region 3: 548 to 600 nm, region 4: 671 to 775.16 nm, region 5: 803.5 to 820.16 nm, and region 6: 821.61 to 839 nm waveband regions of HSI data of the lettuces grown in hydroponic tubs from FDR were identified, and six waveband values 410.33, 522.71, 567.88, 713.51, 811.75, 829.31 nm were determined from these regions. Similarly, for the lettuce grown in the NFT system, 390.9 to 470, 499 to 549, 550 to 599, and 675.98 to 757 nm bandwidth regions were identified, and four waveband values 410.33, 521.38, 571.87, and 713.51 nm were found. For plants, the most useful wavelength ranges for analysis are the visible range in combination with the near-infrared range. In this wavelength range, changes in leaf pigmentation at 400–700 nm and mesophyll cell structure at 700–1300 nm can be detected. However, to detect changes in the water content of a plant, extended ranges are required 1300–2500 nm [44]. In the studies, it was found that the wavelength range of 400–900 nm responded very well to plant characteristics [45]. The ranges 4, 5, and 6 found for hydroponics are consistent with the wavebands of 700 to 709, 780 to 787, and 817 to 821 nm, which were found to be appropriate for the HSI analysis of spinach and bok choy leaves in the studies of Nguyen and Nansen [46]. Two other studies showed that the bandwidth range of 670 to 760 nm provides very accurate prediction models for healthy vegetation [47]. In the previous studies of the authors with four different hydroponically grown cultivars with different N applications, four waveband regions, i.e., 506.33 to 601.11, 670 to 760, 808 to 820, and 821 to 833 nm, were determined to be most appropriate for nutrient-estimation models [16].

Second, using the found multivariate linear regression model coefficients $\beta_{i}$ from the PLSR and PCA by taking the whole range of wavebands of 390.96 to 1008.20 nm and plotting $\beta_{i}$ values against wavebands, six waveband values in the most significant dynamic regions from the plotted data were located for a hydroponic system (Figure 5a) and NFT system (Figure 5b). Six essential HSI reflectance values corresponding to 390.57, 455.19, 542.62, 589.18, 621.17, and 701.44 nm wavebands for a hydroponic system and four HSI values corresponding to 431.42, 551.92, 684.01, and 721.56 nm for NFT system were found. 

Third, VIP-score-based waveband selection was implemented by computing VIP-score values using Equations (7) and (8). From the computed VIP-scores and indices of the wavebands (for a hydroponic system, Figure 5c, and NFT system, Figure 5d), those which exhibited higher than “1” scores were located. The found wavebands were 422.19, 444.62, 551.92, 670.62, 709.49, and 748.44 nm, corresponding to indices 25, 42, 123, 212, 241, and 270 for hydroponically grown lettuces and 419.55, 438.02, 677.32, and 587.84 nm matching with indices of 23, 37, 217, and 150 for lettuces grown in an NFT system. That was subsequently used to select respective HSI reflectance values.

To select an appropriate number of input variables (HSI data corresponding to the found specific wavebands), variance accuracy analysis of the model was performed (Figure 6a–d) for hydroponically and NFT-grown lettuce (data not shown). The analysis performed indicated that six and three input variables for hydroponically and NFT-grown lettuce, respectively, would yield sufficiently high accuracy for the multivariate regression models to estimate nutrient content. 

### 3.4. Comparison of Performance of FDR, PLSR/PCA, and VIP-Score Approach for Estimating Nutrient Content in Lettuce Plants

The quality and accuracy of the chosen wavebands and found models were assessed using the R^2^ and RMSE based on Equation (5). The performances of the multivariate linear regression models for estimating nutrient levels, such as averaged fresh and dried weights in grams), NO_3_^−^, Ca^2+^, K^+^, SPAD, Brix, and pH, were evaluated against the waveband indices found using FDR, PLSR/PCA, and VIP-score approach. The accuracy of the regression models was determined by comparison with data measured in the laboratory. The accuracy of the estimated values of the nutrient levels, including the applied treatment N concentrations, was close $\left(R^{2}=0.91\;\mathrm{to}\;0.99\right)$ to the laboratory-measured data (Figure 7a–e). The comparisons of R^2^ and RMSE values of the studied parameters of different lettuce cultivars grown in hydroponics and the NFT system are shown in Figure 8a–d. The parameters in hydroponics (Figure 8a,b) generally had higher R^2^ and lower RMSE values than those in the NFT system (Figure 8c,d). For example, for lettuces grown in hydroponic tubs, the R^2^ values of dried leaf weight parameter for Black Seeded Simpsons, Parris, Rex RZ, and Tacitus ranged from 0.82 to 0.91, 0.92 to 0.96, 0.55 to 0.99, and 0.73 to 0.95, respectively, whereas in the NFT system, the values varied from 0.51 to 0.54, 0.94 to 0.96, 0.57 to 0.75, and 0.98 to 0.99, respectively. The RMSE values of the dried leaf weight parameter varied for the lettuces grown in hydroponic tubs from 0.33 to 0.47, 0.63 to 0.89, 0.079 to 0.52, and 0.264 to 0.63 g for Black Seeded Simpsons, Parris, Rex RZ, and Tacitus, respectively, while RMSE and R^2^ values for the lettuces grown in the NFT system varied 5.44 to 5.63, 0.66 to 0.81, 1.23 to 1.59, and 0.05 to 0.68 g, respectively. Among the studied cultivars, the found parameter values of Black Seed Simpson in terms of RMSE and R^2^ were most accurate. Among the analyzed parameters, the values of Brix were found to be a best estimate for Black Seeded Simpsons, Parris, Rex RZ, and Tacitus grown in hydroponic tubs, with average R^2^ and RMSE values of 0.99 and 0.20, 0.97 and 0.89, 0.96 and 0.88, and 0.98 and 0.52, respectively, whereas these values were 0.98 and 0.50, 0.94 and 0.76, 0.66 and 0.48, and 0.99 and 0.28, respectively, for the lettuces grown in the NFT system. The regression models using the indices of wavebands from PLSR/PCA and VIP score were significantly better than those with FDR-index models for hydroponic cultivars (Table 5). On the other hand, for the NFT-grown cultivars, FDR-index-based regression models performed much better than the other model indices (Table 6).

Direct comparison with previous studies is not possible because the research hypothesis, objectives, materials, and methods of our study are not the same as those of the other studies. However, in similar studies, highly accurate regression coefficients (R) were obtained for different properties of plants. For example, Sabzi et al. [20] predicted N content in leaves of cucumber plants using HSI and three different regression methods, including a hybrid artificial neural network–particle swarm optimization (ANN-PSO), convolutional neural networks (CNN), and PLSR. The results showed that the PLSR (0.975–0.997) performed slightly better than CNN (0.965–0.985) and ANN-PSO (0.937–0.965). Gao and Xu [48] obtained an optimal PLSR model (0.977) for estimating the soluble solids content of red grapes. In another study [13] conducted using HSI to determine total N content in pepper plants, the PLSR model gave a promising result (0.876). We note that our models are also estimated very accurately, with a mean R^2^ of 0.911 for hydroponics and 0.877 for NFT-grown cultivars. Therefore, our method is consistent with the accuracy of similar models. In our view, this is a beneficial indication that the proposed method can be used to detect nutrient concentration in plants so that farmers can solve problems before they become serious. 

## 4. Conclusions

Four cultivars of leaf lettuce cultured in hydroponic tubs and in an NFT system produced plants with the highest quality and yield when the nutrient solution contained 100–200 ppm N using a commercial 20−20−20 fertilizer. When plants were grown at higher N concentrations (300–350 ppm), some plants developed leaf-edge burns, which reduce the marketability of the vegetable. The HSI technique is reliable and highly accurate for estimating the nutrient levels of the lettuce cultivars in vivo with correctly selected waveband values using multivariate regression models. The nutrient concentration of the lettuce leaves in some industrial applications (restaurants) is important since some restaurant chains impose regulatory requirements regarding the consistency of nutrient content. Hence, estimating the lettuce nutrient concentration rapidly is important. The most appropriate wavebands for selecting the measured HSI data for nutrient estimation are 390.57, 455.19, 542.62, 589.18, 621.17, and 701.44 nm for hydroponic cultivars and 419.55, 438.02, and 677.32 nm for NFT-circulation-system-grown lettuces. The results of the study show that there is a direct correlation between the HSI of the applied nutrient solutions and the lettuce leaves. The developed methodological approach with an HSI technique has sound application potential not only for lettuce or other crops grown in greenhouses but also in the field. Moreover, the proposed methodology with an HSI technique can be also applied for the evaluation of lettuce leaves before harvesting and grading of the harvested product before sales. 

The future work will be dedicated to study the growth dynamics of lettuce and other plants in the fields using the HSI technique with applications of machine learning algorithms for estimation and predictive model development. We will employ artificial neural networks (ANN), support vector regression (SVR), and random forest (RF) algorithms to estimate and develop predictive models. Moreover, the HSI imaging technique will be applied to the same lettuce cultivars studied here by using different types of light with different light intensities plus different exposure conditions. 

## Figures and Tables

**Figure 1 sensors-22-08158-f001:**
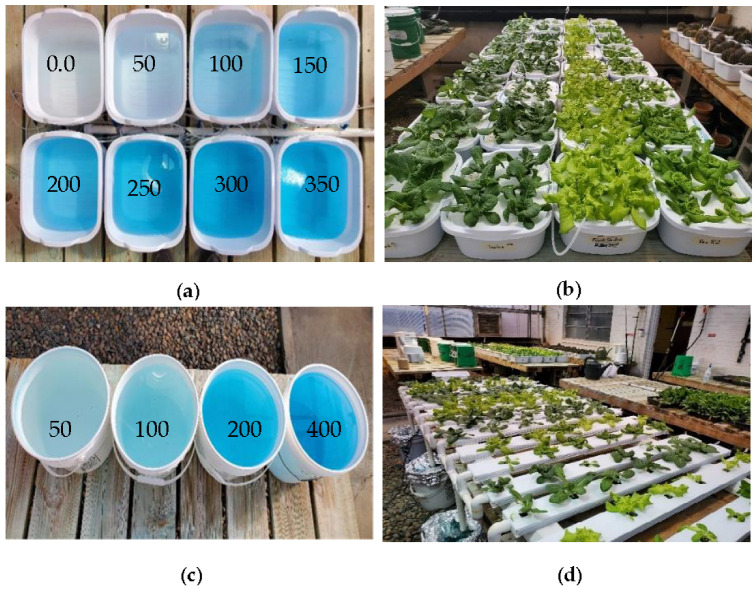
(**a**) Nutrient solutions in different concentrations, ppm; (**b**) four lettuce cultivars, including Parris, Tacitus, Black Seed Simpson, and Rex (left to right), grown in hydroponic tubs; (**c**) nutrient solutions in different concentrations, ppm; (**d**) four lettuce cultivars, including Parris, Black Seed Simpson, Rex, and Tacitus (left to right), grown in the NFT system for ten days.

**Figure 2 sensors-22-08158-f002:**
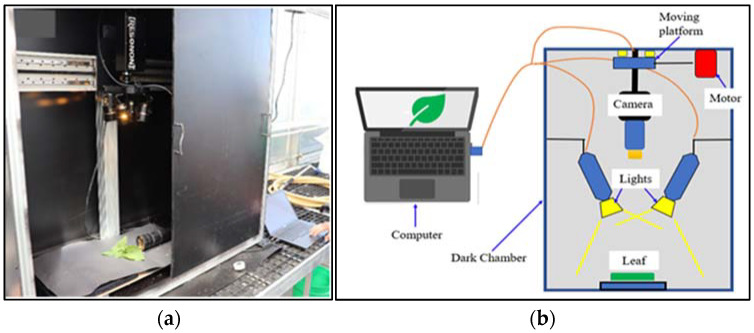
Hyperspectral camera: (**a**) HSI system (camera: Pika XC2, Resonon Inc); (**b**) schematic representation of the HSI scanning system.

**Figure 3 sensors-22-08158-f003:**
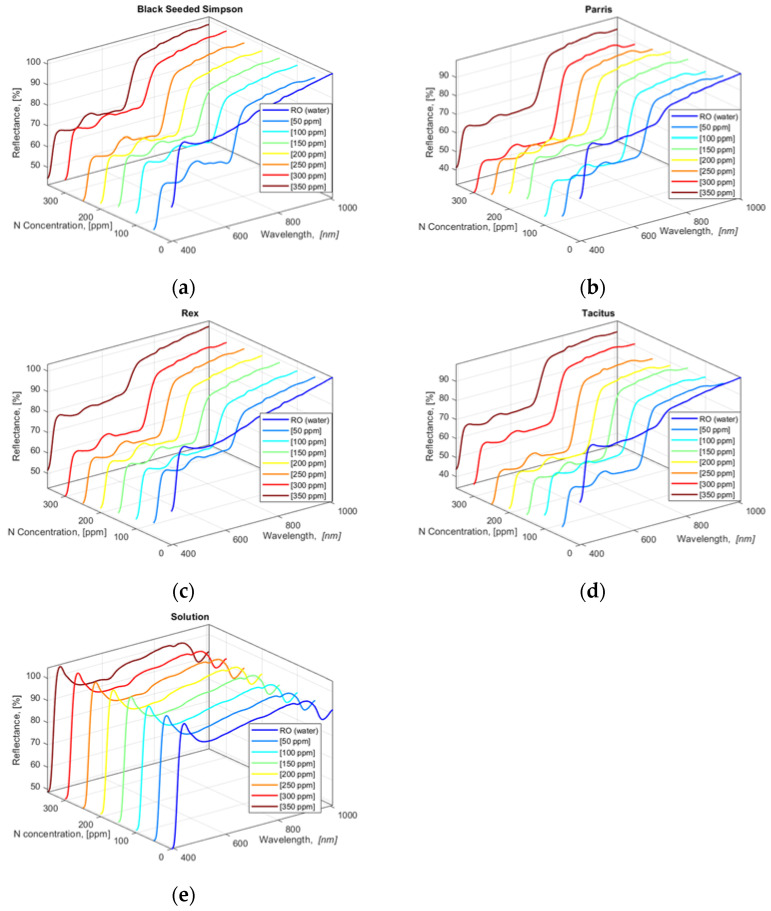
Averaged and smoothed reflectance values of HSI pixel values of four freshly cut lettuce cultivars: (**a**) Black Seeded Simpson, (**b**) Parris, (**c**) Rex RZ, and (**d**) Tacitus and (**e**) solution in hydroponic tubs with different N concentrations.

**Figure 4 sensors-22-08158-f004:**
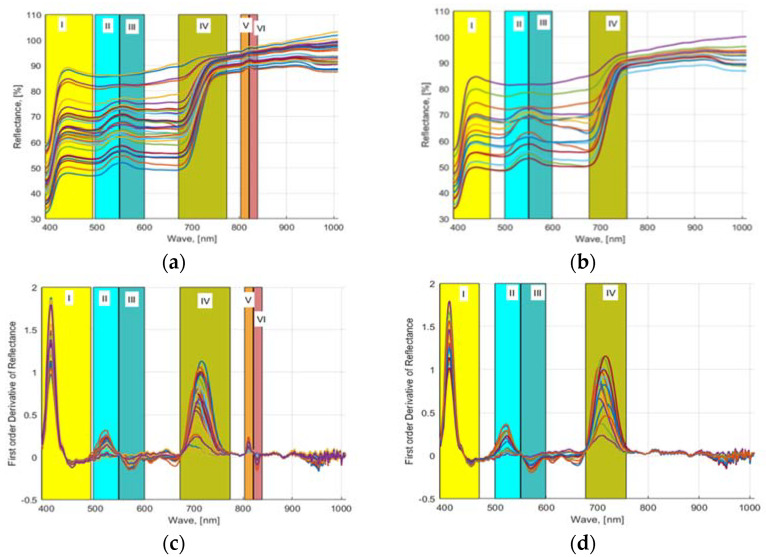
NDS and FDR indices: (**a**,**b**) HSI reflectance values from 32 freshly cut leaves of four lettuce cultivars (Black Seeded Simpsons, Parris, Rex RX, Tacitus) grown in hydroponic tubs and NFT system with different N concentrations and (**c**,**d**) their computed first-order derivatives.

**Figure 5 sensors-22-08158-f005:**
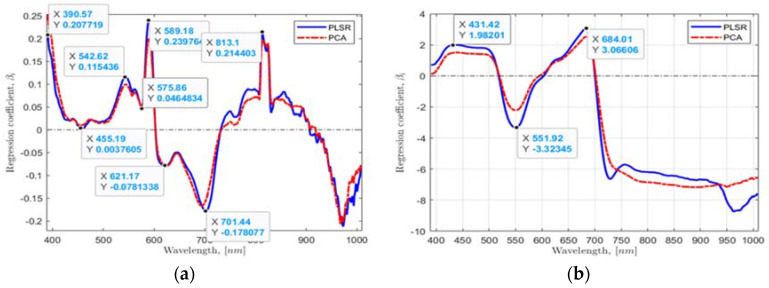
The found regression coefficients using: (**a**,**b**) PLSR and PCA, and (**c**,**d**) VIP-scores of PLSR and PCA approaches for the hydroponic tubs and NFT system, respectively.

**Figure 6 sensors-22-08158-f006:**
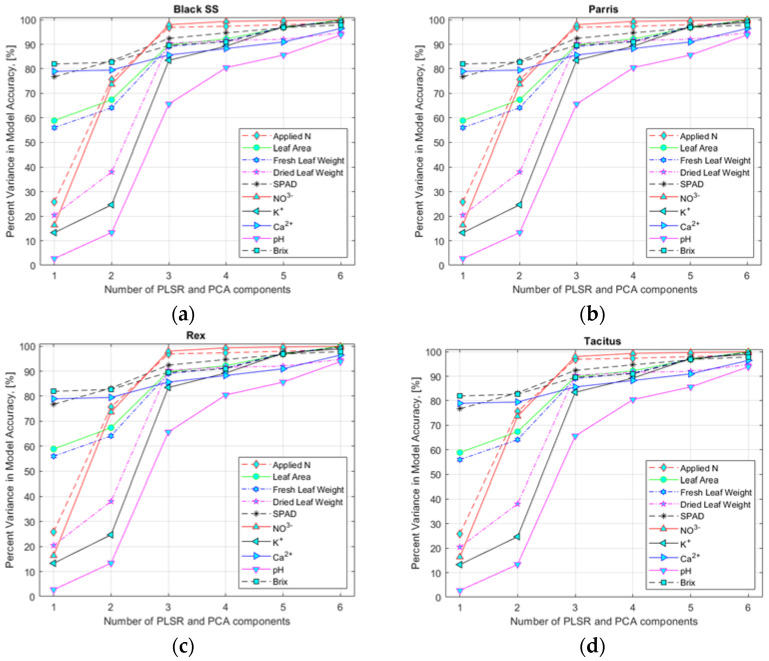
Selection of the number of predictor variable components using the PLSR and PCA models for lettuce cultivars: (**a**) Black Seeded Simpsons, (**b**) Parris, (**c**) Rex RZ, and (**d**) Tacitus grown in hydroponic tubs with different N concentrations.

**Figure 7 sensors-22-08158-f007:**
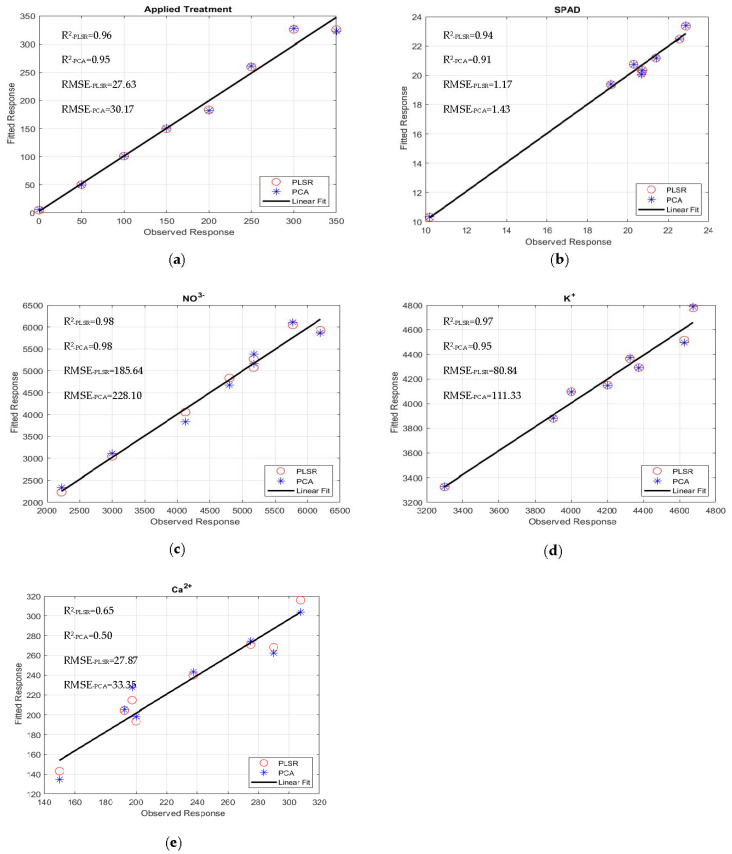
Observed and fitted (predicted) response (Black Seeded Simpsons): (**a**) applied treatment, (**b**) SPAD, (**c**) NO_3_^−^, (**d**) K^+^, and (**e**) Ca^2+.^

**Figure 8 sensors-22-08158-f008:**
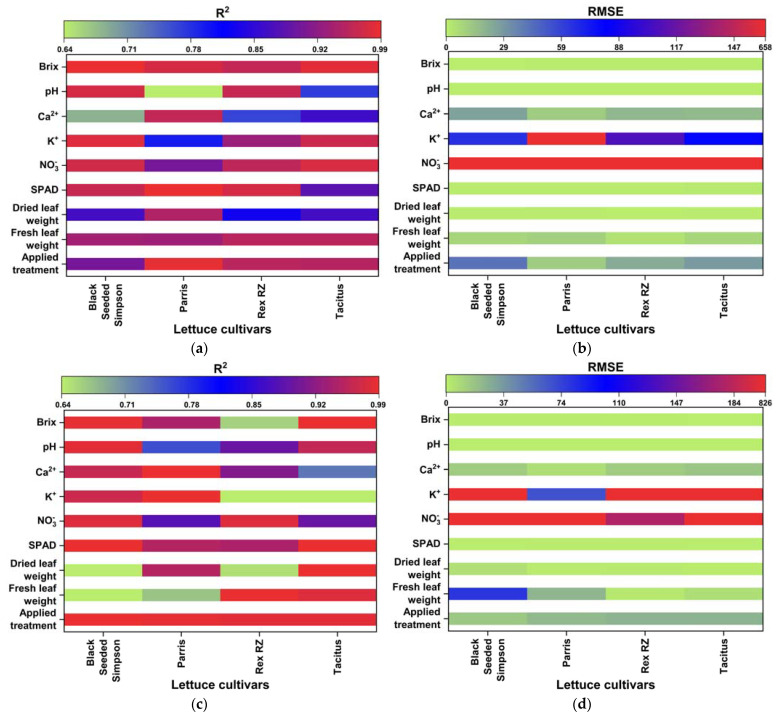
R^2^ and RMSE values of the investigated parameters of different lettuce cultivars grown in hydroponics (**a**,**b**) and in the NFT system (**c**,**d**).

**Table 1 sensors-22-08158-t001:** Influence of nutrient concentrations on the growth of four different lettuce cultivars in hydroponic culture.

Treatment (ppm N)	Fresh Weight (g/Plant)	Dry Weight (g/Plant)	Visual Quality *	Leaf-Edge Burns **	SPAD
Black Seeded Simpson					
0	4.8 ± 1.0	-	2.0	5.0	10.2 ± 1.5
50	69.0 ± 16.7	3.2 ± 1.2	5.0	5.0	19.2 ± 2.2
100	80.8 ± 34.9	3.6 ± 2.3	5.0	5.0	20.7 ± 2.2
150	59.0 ± 23.2	2.6 ± 0.9	5.0	5.0	22.6 ± 6.1
200	72.5 ± 18.3	2.9 ± 0.8	5.0	5.0	20.3 ± 3.5
250	104.8 ± 21.7	5.1 ± 1.0	5.0	5.0	20.7 ± 3.5
300	78.0 ± 36.2	4.1 ± 2.7	5.0	4.5	21.4 ± 1.5
350	68.0 ± 5.3	3.5 ± 0.2	5.0	4.5	22.9 ± 4.9
Parris Island					
0	5.5 ± 3.1	-	2.0	5.0	30.3 ± 6.2
50	59.3 ± 18.8	8.1 ± 3.2	5.0	5.0	39.6 ± 2.7
100	79.8 ± 31.0	10.9 ± 5.3	5.0	5.0	42.6 ± 2.7
150	116.8 ± 21.9	4.7 ± 1.2	5.0	5.0	39.6 ± 3.5
200	111.5 ± 30.9	5.0 ± 2.2	5.0	5.0	40.4 ± 2.7
250	121.0 ± 51.0	4.3 ± 2.2	5.0	5.0	42.5 ± 1.4
300	98.3 ± 13.8	4.4 ± 0.9	5.0	5.0	43.6 ± 6.5
350	117.3 ± 29.7	5.7 ± 1.9	5.0	5.0	45.8 ± 3.6
Rex RZ					
0	5.5 ± 3.7	-	2.5	5.0	23.0 ± 2.2
50	54.0 ± 3.7	2.3 ± 0.3	5.0	5.0	30.9 ± 2.7
100	76.0 ± 20.3	3.8 ± 1.7	5.0	5.0	29.5 ± 1.4
150	69.0 ± 13.6	2.8 ± 0.8	5.0	5.0	31.3 ± 2.9
200	69.0 ± 13.1	3.5 ± 2.0	5.0	5.0	30.6 ± 1.1
250	70.3 ± 5.7	2.8 ± 0.3	5.0	5.0	30.4 ± 4.5
300	62.5 ± 24.4	2.5 ± 1.1	5.0	5.0	32.2 ± 3.5
350	72.5 ± 36.5	3.9 ± 2.4	5.0	4.5	28.7 ± 1.7
Tacitus					
0	5.0 ± 2.2	-	2.5	5.0	36.6 ± 2.1
50	59.8 ± 14.3	2.8 ± 0.9	5.0	5.0	44.3 ± 1.6
100	73.8 ± 36.3	3.0 ± 2.1	5.0	5.0	45.3 ± 2.7
150	97.3 ± 18.3	4.3 ± 2.4	5.0	5.0	44.7 ± 2.3
200	105.8 ± 26.3	4.0 ± 1.2	5.0	5.0	46.5 ± 1.7
250	84.0 ± 29.1	3.2 ± 1.8	5.0	5.0	45.9 ± 3.1
300	102.3 ± 35.5	4.6 ± 1.9	5.0	5.0	45.4 ± 3.5
350	94.8 ± 32.9	5.1 ± 3.8	5.0	4.0	47.7 ± 2.6

* Ratings of quality from 1 (very poor) to 5 (excellent). ** Ratings of leaf-edge burns: from 1 (severe) to 5 (no symptom).

**Table 2 sensors-22-08158-t002:** Influence of nutrient concentrations on tissue mineral composition, pH, and solids content of four different lettuce cultivars grown hydroponic tub cultures.

Treatment (ppm N)	Tissue NO_3_^−^ (ppm)	Tissue K^+^ (ppm)	Tissue Ca^2+^ (ppm)	Tissue pH	Brix (%)
Black Seeded Simpson					
0	3000 ± 1074	3300 ± 1046	308 ± 113	6.2 ± 0.7	12.5 ± 0.5
50	2225 ± 763	4325 ± 585	333 ± 48	5.9 ± 0.1	7.4 ± 0.3
100	4125 ± 690	4625 ± 171	223 ± 95	5.9 ± 0.1	6.9 ± 1.3
150	5175 ± 1315	3900 ± 726	243 ± 40	5.9 ± 0.2	6.5 ± 1.7
200	4800 ± 663	4000 ± 1078	265 ± 26	5.9 ± 0.3	6.7 ± 0.9
250	5175 ± 1209	4375 ± 624	240 ± 16	6.0 ± 0.1	7.8 ± 0.7
300	6200 ± 683	4200 ± 825	248 ± 25	6.0 ± 0.3	6.0 ± 0.7
350	5775 ± 785	6025 ± 606	308 ± 115	6.1 ± 0.3	7.8 ± 1.5
Parris Island					
0	1650 ± 173	5300 ± 956	150 ± 25	6.1 ± 0.3	15.3 ± 4.9
50	2050 ± 412	5250 ± 834	238 ± 47	6.2 ± 0.2	7.8 ± 1.2
100	4450 ± 1034	5750 ± 1622	193 ± 13	6.0 ± 0.1	7.3 ± 0.8
150	5025 ± 299	4500 ± 510	275 ± 29	5.9 ± 0.1	6.3 ± 0.9
200	5975 ± 866	4050 ± 433	200 ± 29	6.0 ± 0.1	5.9 ± 0.9
250	5875 ± 1028	4800 ± 993	308 ± 39	5.8 ± 0.2	6.7 ± 1.0
300	5333 ± 306	4200 ± 346	290 ± 26	5.9 ± 0.1	7.3 ± 1.1
350	6675 ± 834	5475 ± 525	198 ± 35	6.1 ± 0.1	8.2 ± 0.2
Rex RZ					
0	3325 ± 754	4175 ± 1021	318 ± 99	7.2 ± 1.0	18.7 ± 2.5
50	2138 ± 221	5350 ± 1047	335 ± 34	6.3 ± 0.3	7.6 ± 1.2
100	4825 ± 465	5050 ± 755	298 ± 34	5.9 ± 0.1	7.2 ± 1.0
150	5775 ± 556	4925 ± 499	233 ± 40	5.9 ± 0.1	6.9 ± 0.9
200	6825 ± 704	5500 ± 990	310 ± 83	6.0 ± 0.1	6.7 ± 0.7
250	7850 ± 387	5425 ± 1034	278 ± 82	5.9 ± 0.1	6.5 ± 0.7
300	7375 ± 574	4975 ± 465	255 ± 53	6.0 ± 0.4	6.7 ± 1.1
350	7450 ± 1162	4725 ± 411	213 ± 38	6.0 ± 0.1	7.0 ± 0.8
Tacitus					
0	2425 ± 435	4850 ± 881	185 ± 37	6.1 ± 0.5	19.3 ± 1.4
50	1625 ± 435	5250 ± 881	318 ± 66	6.2 ± 0.4	8.5 ± 0.8
100	3375 ± 403	4700 ± 707	283 ± 59	6.0 ± 0.2	7.5 ± 1.6
150	4800 ± 1197	4250 ± 676	248 ± 64	5.9 ± 0.1	7.0 ± 1.3
200	4575 ± 465	4450 ± 526	323 ± 114	5.9 ± 0.1	6.9 ± 1.0
250	5475 ± 525	4600 ± 497	298 ± 69	6.2 ± 0.2	6.8 ± 1.0
300	5600 ± 852	4150 ± 480	295 ± 72	6.2 ± 0.1	7.6 ± 0.6
350	6650 ± 575	4600 ± 825	298 ± 56	6.1 ± 0.3	9.2 ± 0.3

**Table 3 sensors-22-08158-t003:** Influence of nutrient concentrations on tissue mineral composition, pH, and solids content of four different lettuce cultivars grown in the NFT system.

Treatment (ppm N)	Fresh Weight (g/Plant)	Dry Weight (g/Plant)	Visual Quality *	Leaf Edge Burns **	SPAD	Tissue NO_3_^−^ (ppm)	Tissue K^+^ (ppm)	Tissue Ca^2+^ (ppm)	Tissue pH	Brix (%)
Black Seeded Simpson										
50	82.5 ± 35.1	4.4 ± 2.3	4.0	5.0	20.1 ± 4.2	2925 ± 670	3325 ± 613	218 ± 30	5.9 ± 0.1	6.6 ± 0.9
100	198.5 ± 54.6	15.0 ± 10.0	4.5	4.5	21.4 ± 2.0	4850 ± 252	4025 ± 655	290 ± 20	5.9 ± 0.1	6.7 ± 0.8
200	254.0 ± 46.0	18.7 ± 9.5	5.0	4.0	24.7 ± 3.8	5925 ± 492	3975 ± 842	250 ± 26	5.9 ± 0.2	8.0 ± 0.7
400	65.0 ± 10.8	7.8 ± 1.4	2.0	3.0	32.2 ± 5.1	8300 ± 1214	5925 ± 896	147 ± 44	5.6 ± 0.2	14.1 ± 0.7
Parris Island										
50	128.8 ± 22.7	2.8 ± 1.2	4.5	5.0	38.8 ± 3.6	2475 ± 826	4000 ± 1707	218 ± 46	6.3 ± 0.4	6.8 ± 1.0
100	170.0 ± 37.1	3.9 ± 1.7	5.0	5.0	40.1 ± 4.4	5550 ± 404	4550 ± 819	238 ± 38	6.0 ± 0.1	6.0 ± 1.2
200	141.0 ± 31.1	8.4 ± 4.7	4.3	4.8	45.7 ± 2.4	6675 ± 1053	4650 ± 574	208 ± 40	5.9 ± 0.1	6.8 ± 1.0
400	86.0 ± 11.9	7.3 ± 0.1	3.8	4.3	46.2 ± 4.6	5825 ± 685	8000 ± 735	325 ± 159	6.3 ± 0.4	11.7 ± 2.5
Rex RZ										
50	110.3 ± 31.5	7.9 ± 3.2	4.5	5.0	29.2 ± 1.8	6725 ± 763	3700 ± 816	240 ± 25	5.9 ± 0.1	6.1 ± 0.8
100	145.8 ± 31.5	11.1 ± 6.5	5.0	5.0	28.9 ± 2.0	7025 ± 378	4575 ± 222	263 ± 19	5.8 ± 0.1	5.9 ± 0.6
200	125.8 ± 38.2	6.8 ± 4.1	4.8	4.8	29.0 ± 2.4	6100 ± 956	5125 ± 655	260 ± 22	5.9 ± 0.1	7.1 ± 2.7
400	92.0 ± 25.3	7.0 ± 3.1	4.0	4.0	35.7 ± 6.8	4750 ± 1147	4450 ± 2089	176 ± 36	5.8 ± 0.1	7.2 ± 0.4
Tacitus										
50	88.3 ± 34.5	2.2 ± 3.8	4.8	5.0	42.6 ± 1.3	2875 ± 427	3425 ± 403	210 ± 8	6,1 ± 0.2	8.5 ± 0.3
100	123.5 ± 58.3	12.6 ± 9.9	5.0	5.0	41.4 ± 3.7	6750 ± 569	5450 ± 968	210 ± 55	5.9 ± 0.1	5.9 ± 0.8
200	158.5 ± 42.2	15.4 ± 9.2	4.3	4.5	42.8 ± 1.4	6800 ± 860	5025 ± 299	258 ± 77	5.8 ± 0.2	6.9 ± 1.2
400	74.5 ± 21.4	7.0 ± 2.3	3.0	3.8	48.9 ± 5.5	3550 ± 881	5400 ± 1520	195 ± 45	5.8 ± 0.0	22.6 ± 2.7

* Ratings of quality from 1 (very poor) to 5 (excellent). ** Ratings of leaf-edge burns: from 1 (severe) to 5 (no symptom).

**Table 4 sensors-22-08158-t004:** Cross-correlation analysis of HSI data of lettuce leaves and applied N concentration solutions.

Treatment (ppm N)	$\mathrm{Correlation}\;\mathrm{Value}\text{: }r_{corr}$			
	Black Seeded Simpson	Parris	Rex RZ	Tacitus
0	0.979	0.975	0.981	0.979
50	0.821	0.908	0.920	0.873
100	0.893	0.855	0.904	0.925
150	0.912	0.919	0.876	0.873
200	0.879	0.888	0.861	0.885
250	0.851	0.872	0.879	0.843
300	0.918	0.844	0.864	0.925
350	0.881	0.936	0.952	0.941

**Table 5 sensors-22-08158-t005:** Fitted model accuracy (R^2^, RMSE) of models using waveband indices found from FDR, PLSR/PCA, and VIP score of PLSR/PCA for hydroponically grown lettuce cultivars.

Type	FDR Index		PLSR/PCA Index		VIP-Score Index	
	R^2^	RMSE	R^2^	RMSE	R^2^	RMSE
Black Seeded Simpson						
Applied Treatment	0.82	62.01	0.96	27.63	0.92	36.78
Fresh Leaf Weight	**0.97**	**5.93**	0.89	10.98	0.94	7.68
Dried Leaf Weight	**0.91**	**0.33**	0.82	0.47	0.86	0.38
SPAD	0.95	1.05	0.94	1.171	**0.99**	**0.03**
NO_3_^−^	0.93	437.19	0.98	185.63	**0.99**	**111.51**
K^+^	0.97	83.69	0.97	80.84	**0.99**	**30.02**
Ca^2+^	0.75	23.76	0.65	27.87	**0.65**	**25.42**
pH	0.95	0.03	0.98	0.02	**0.98**	**0.016**
Brix	0.98	0.36	0.99	0.21	**0.99**	**0.03**
Parris						
Applied Treatment	0.97	22.43	0.99	11.91	**0.99**	**0.73**
Fresh Leaf Weight	0.92	13.05	0.88	16.08	**0.99**	**0.76**
Dried Leaf Weight	0.92	0.89	**0.96**	**0.63**	0.95	0.63
SPAD	0.99	0.44	0.98	0.731	**0.99**	**0.26**
NO_3_^−^	0.87	788.66	0.88	760.39	**0.95**	**419.61**
K^+^	0.63	449.38	**0.95**	**160.99**	0.80	302.20
Ca^2+^	**0.99**	**5.51**	0.94	15.57	0.94	14.35
pH	0.58	0.09	0.49	0.098	**0.85**	**0.05**
Brix	0.97	0.92	0.97	0.90	**0.97**	**0.85**
Rex RZ						
Applied Treatment	**0.99**	**8.04**	0.99	12.437	0.87	47.79
Fresh Leaf Weight	0.87	9.91	0.99	0.840	**0.99**	**0.27**
Dried Leaf Weight	0.55	0.52	**0.99**	**0.079**	0.94	0.17
SPAD	0.98	0.41	**0.99**	**0.061**	0.94	0.72
NO_3_^−^	0.88	854.39	0.99	125.99	**0.99**	**97.51**
K^+^	0.80	227.47	**0.99**	**48.52**	0.99	48.55
Ca^2+^	0.35	41.35	**0.99**	**3.196**	0.95	10.79
pH	0.90	0.16	0.99	0.019	**0.99**	**0.016**
Brix	0.92	1.42	**0.99**	**0.345**	0.96	0.87
Tacitus						
Applied Treatment	0.87	52.88	**0.99**	**15.50**	0.98	16.90
Fresh Leaf Weight	0.90	12.45	**0.98**	**4.911**	0.97	6.403
Dried Leaf Weight	0.73	0.63	**0.95**	**0.264**	0.91	0.33
SPAD	0.82	1.724	0.91	1.22	**0.91**	**1.09**
NO_3_^−^	0.96	361.98	0.97	311.36	**0.99**	**152.50**
K^+^	0.97	75.04	0.96	85.37	**0.96**	**78.75**
Ca^2+^	0.71	28.47	**0.96**	**9.95**	0.91	14.56
pH	0.83	0.057	0.74	0.07	0.74	0.06
Brix	0.97	0.76	0.98	0.61	**0.99**	**0.20**

**Table 6 sensors-22-08158-t006:** Fitted model accuracy (R^2^, RMSE) of models found from RSI, NDSI, PLSR/PCA, and VIP score of PLSR/PCA for NFT-circular-system-grown lettuce cultivars.

Type	FDR Index		PLSR/PCA Index		VIP-Score Index	
	R^2^	RMSE	R^2^	RMSE	R^2^	RMSE
Black Seeded Simpson						
Applied Treatment	0.99	18.04	**0.99**	**15.19**	0.99	15.494
Fresh Leaf Weight	0.48	80.13	**0.51**	**78.37**	0.50	78.548
Dried Leaf Weight	0.51	5.63	**0.54**	**5.44**	0.54	5.45
SPAD	0.99	0.37	**0.99**	**0.27**	0.99	0.28
NO_3_^−^	0.97	425.41	**0.98**	**363.325**	0.98	369.58
K^+^	0.97	227.76	0.96	253.99	**0.96**	**251.38**
Ca^2+^	0.96	14.93	**0.96**	**14.19**	0.96	14.26
pH	**0.98**	**0.02**	**0.98**	**0.02**	**0.98**	**0.02**
Brix	**0.99**	**0.47**	0.98	0.52	0.98	0.51
Parris						
Applied Treatment	**0.99**	**20.56**	0.98	22.86	0.98	24.71
Fresh Leaf Weight	**0.69**	**23.71**	0.66	24.80	0.66	24.77
Dried Leaf Weight	**0.96**	**0.66**	0.94	0.75	0.94	0.81
SPAD	**0.96**	**0.95**	0.94	1.06	0.94	1.13
NO_3_^−^	0.86	830.42	0.88	775.79	0.89	734.28
K^+^	**0.99**	**57.30**	0.99	66.37	0.99	77.89
Ca^2+^	**0.99**	**6.43**	0.99	6.66	0.99	6.59
pH	0.74	0.118	0.76	0.114	0.77	0.11
Brix	**0.94**	**0.754**	0.94	0.757	0.94	0.76
Rex RZ						
Applied Treatment	**0.98**	**23.34**	0.98	25.45	0.98	25.88
Fresh Leaf Weight	0.97	4.37	0.998	1.1830	**0.99**	**1.07**
Dried Leaf Weight	**0.75**	**1.23**	0.63	1.49	0.57	1.59
SPAD	0.92	1.15	0.94	0.99	**0.96**	**0.83**
NO_3_^−^	**0.99**	**128.67**	0.97	186.55	0.97	218.64
K^+^	0.45	526.96	0.62	456.71	**0.72**	**388.02**
Ca^2+^	0.87	17.27	0.91	14.13	**0.95**	**11.31**
pH	**0.93**	**0.007**	0.88	0.01	0.86	0.011
Brix	0.69	0.46	0.64	0.49	0.65	0.49
Tacitus						
Applied Treatment	**0.99**	**18.67**	0.98	25.99	0.97	29.16
Fresh Leaf Weight	0.96	9.60	0.98	5.71	0.99	4.25
Dried Leaf Weight	**0.99**	**0.05**	0.99	0.49	0.98	0.68
SPAD	**0.99**	**0.16**	0.99	0.17	0.99	0.18
NO_3_^−^	**0.93**	**671.45**	0.88	862.95	0.86	938.71
K^+^	**0.61**	**731.12**	0.51	812.18	0.48	841.57
Ca^2+^	0.67	19.12	0.74	16.97	**0.77**	**15.91**
pH	**0.98**	**0.02**	0.95	0.028	0.94	0.03
Brix	**0.99**	**0.22**	0.99	0.297	0.99	0.32

## Data Availability

Not applicable.

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
