# Peer review of "Hyperspectral Image Data and Waveband Indexing Methods to Estimate Nutrient Concentration on Lettuce (Lactuca sativa L.) Cultivars"

_sensors, 2022, doi:10.3390/s22218158_

Round 1
Reviewer 1 Report
1. The author analyzed the concentration on potassium, phosphorus and nitrogen in hyperspectral reflectance values, kindly compute this work with first order and second order differential equations.
2. Kindly mention on example calculation, how the concentration or attains for SPAD.
3. Mention about the details of dataset description or link for the dataset (please put as references)
4. Figure 5 resolution is very poor. kindly change the resolution of the image.
5. Kindly include the comparative analysis of the R square and RMSE. (Like charts)
6. Include the future work in the conclusion.
7. Why PLSR & PCA why not SVR kindly compare the same, how its performed.
8. Elaborate the novelty of your work.
9. Simply new datasets won't make a good sense. kindly update some new innovation in your paper.
Author Response
Thank you so much for your time and efforts. Our answers are in the attached file.
Here are also our answers to your comments:
- The author analyzed the concentration on potassium, phosphorus and nitrogen in hyperspectral reflectance values, kindly compute this work with first order and second order differential equations.
Answer: First order derivative values are computed to find waveband values that are shown in Figure 4 c and d.
- Kindly mention on example calculation, how the concentration or attains for SPAD.
Answer: This question is addressed in section 2.1.
- Mention about the details of dataset description or link for the dataset (please put as references) –
Answer: One MS Excel table of the dataset is added as an appendix.
- Figure 5 resolution is very poor. kindly change the resolution of the image.
Answer: The resolution of Figure 5 is corrected.
- Kindly include the comparative analysis of the R square and RMSE. (Like charts)
Answer: Figure 8 is added to the manuscript.
- Include the future work in the conclusion.
Answer: Future work plan is added in the conclusion.
- Why PLSR & PCA why not SVR kindly compare the same, how its performed.
Answer: The motivation for applying PLSR/PCA over machine learning tools is added along with some discussions on comparative analysis studies of machine learning tools and PLSR/PCA.
- Elaborate the novelty of your work. –
Answer: The novelty features of the work are elaborated on at the end of the Introduction.
- Simply new datasets won't make good sense. kindly update some new innovation in your paper.
Answer: As explained in the manuscript, the novelty of our study is the developed methodology to estimate the growth dynamics of lettuce cultivars using HSI data and three different waveband indexing methods. Moreover, another novelty feature of the study is that a correlation between HSI scanning of the nutrient solution versus HSI scanning of lettuce leaves was the first time studied and a good correlation between nutrient concentrations in the solution and the leaves was found.

Reviewer 2 Report
My report is available in the attached file.

Author Response
Thank you for your time and efforts to review our paper.
Here are our answers to your comments (also given in the attached file):
- The manuscript "sensors-1939682" deals with the “Hyperspectral image data and waveband indexing methods to estimate nutrient concentration on lettuce (Lactuca sativa L.) cultivars”. The topic is interesting and relevant to journal. The English grammar is well.
Answer: We appreciate Reviewer’s constructive comments regarding this study and the presented results.
- Abstract A sentence about the importance of the work should be added to the beginning of the paragraph. (Line 13)
Answer: One statement on the importance of lettuces as a food is included at the beginning of the abstract.
- More numerical results should be included in the abstract. (Line 23)
Answer: Numerical results are added to the abstract.
- There are too many keywords, the ones that will reflect the essence of the work should be chosen.
Answer: Several keywords are removed and substituted with one keyword.
- Introduction suffers from lack of motivations and innovations. It should be expanded to include a clear discussion of current problems. The difference from previous studies should be emphasized more clearly. (Line 93)
Answer: The introduction is expanded with discussions of several literature sources and underlining the motivation and analyzing missing gaps. Also, the novelty features of the study are added.
- Material and Methods Briefly explain why PCA was chosen as the feature extraction model. (Line 214)
Answer: The motivation for applying PLSR/PCA is added along with some discussions on comparative analysis studies of machine learning tools and PLSR/PCA.
- The detailed information on the instruments used must be given including the model, manufacturer, name of company, place where the company located, etc.
Answer: Detailed information of the used instrumentation and sensors are included in sections 2.1 and 2.3.
- Please re-write the Statistical analysis to explain the models, factors, treatments that used.
Answer: Some explanations are added.
- Why did you choose FDR, PLS/PCA and VIP-score as predictors, could not one or two machine learning algorithms be used with them? What are the advantages of used models?
Answer: The motivation for applying PLSR/PCA over machine learning tools is added along with some discussions on comparative analysis studies of machine learning tools and PLSR/PCA. The importance of FDR and VIP-score approaches to locate the most appropriate indexes is elaborated.
- Check all formulas in the document. They are not formatted. Define the used symbols clearly and numerate all equations that appear.
Answer: All formulas are reformatted.
- Results and Discussions. Results and Discussion; author should compare the finding of present study with previous study and justify for more clarity.
Answer: Comparative analysis and discussion of our findings and results of other studies are highlighted in Results and Discussion section of the paper.
- Authors should add separate section regarding outlook and specific comment point wise based on their study.
Answer: Separate section 3.4 is added that highlights the comparison of our results with other studies.
- R^2 values can be shared on Figure 7.
Answer: All R^2 and RMSE values are added to Figure 7.
- In this section, the importance of estimate nutrient concentration on lettuce in industrial applications is not mentioned. If appropriate, such a sentence should be added.
Answer: This point is added in the conclusions section.
Thank you.

Round 2
Reviewer 1 Report
Line number 465 : comparison parameter is not clear.
Lack of clarity in SPAD computation.
Author not answered for the query, related to comparison of this work with first and second order differential equations.
Why PLSR & PCA why not SVR kindly compare the same, how its performed
Author Response
Dear Reviewer,
We really appreciate your thorough review and provided comments and remarks.
We have revised and edited the whole manuscript content. Most extensively edited parts are highlighted to address your concerns and comments.
Here are our responses to your posed comments:
Comment 1: Line number 465: comparison parameter is not clear.
Answer: Thank you for spotting this missed point. We have corrected this point – see the corrected text: “The comparisons of R2 and RMSE values of…”. Moreover, we have edited the contexts – see lines 476-493.
Comment 2: Lack of clarity in SPAD computation.
Answer: Thanks for the comment. We have included more explanations about SPAD computation – see lines 62-69, 171-174. Since our employed SPAD measurement sensor is a simple device that gives a SPAD value right away during the measurement. We have also included one literature source that gives a detailed explanation of SPAD readings and calculations.
Comment 3: Author not answered for the query, related to comparison of this work with first and second order differential equations.
Answer: We have included the normalized difference spectral (NDR) of reflectance values and first-order derivative reflectance (FDR) values – Figure 4a vs. 4c, and 4b vs. 4d. We have found in our studies that the found indices of the reflectance values from FDR were highly accurate as found in other studies [16, 26], as well. See – 237-242, 401-410, 466-497.
Comment 4: Why PLSR & PCA why not SVR kindly compare the same, how its performed.
Answer: We highly respect the reviewer's comments and suggestions. It would have been interesting to explore this aspect. However, our aim was to develop models from data from hyperspectral images (HSI) used to estimate the nutrient content of leaf tissues using principal component analysis (PCA), partial least squares regression (PLSR), multivariate regression, and variable importance projection (VIP) methods. See lines 468-513.
We mentioned in the article that future work will be devoted to the study of the growth dynamics of lettuce and other crops in the fields, using the HSI technique with applications of machine learning algorithms, including artificial neural networks (ANN), support vector regression (SVR), random forest (RF) to estimate and develop predictive models. We very much appreciate this helpful comment. Your suggestions will be very useful for the study we plan to conduct in the future.
Thank you.
Sincerely,
Sulaymon Eshkabilov
On the behalf of authors.

Reviewer 2 Report
The requested revisions have been completed. The manuscript is acceptable as it is.
Author Response
Dear Reviewer,
We really appreciate your efforts and positive assessment of our revised manuscript.
Thank you.
Sincerely,
Sulaymon Eshkabilov
On the behalf of authors.
